# Leveraging Prior Experience: An Expandable Auxiliary Knowledge Base for Text-to-SQL

## Abstract

Large Language Models (LLMs) exhibit impressive problem-solving skills across many tasks, but they still underperform compared to humans in various downstream applications, such as text-to-SQL. On the BIRD benchmark leaderboard, human performance achieves an accuracy of 92.96%, whereas the top-performing method reaches only 72.39%. Notably, these state-of-the-art (SoTA) methods predominantly rely on in-context learning to simulate human-like reasoning. However, they overlook a critical human skill: continual learning. Inspired by the educational practice of maintaining mistake notebooks during our formative years, we propose LPE-SQL (**L**everaging **P**rior **E**xperience: An Expandable Auxiliary Knowledge Base for Text-to-**SQL**), a novel framework designed to augment LLMs by enabling continual learning without requiring parameter fine-tuning. LPE-SQL consists of four modules that **i)** retrieve relevant entries, **ii)** efficient sql generation, **iii)** generate the final result through a cross-consistency mechanism and **iv)** log successful and failed tasks along with their reasoning processes or reflection-generated tips. Importantly, the core module of LPE-SQL is the fourth one, while the other modules employ foundational methods, allowing LPE-SQL to be easily integrated with SoTA technologies to further enhance performance. Our experimental results demonstrate that this continual learning approach yields substantial performance gains, with the smaller Llama-3.1-70B model with surpassing the performance of the larger Llama-3.1-405B model using SoTA methods.

## 1 Introduction

Text-to-SQL, the task of converting natural language queries into structured SQL commands, has garnered significant attention due to its potential to simplify database interactions. Recently, in-context learning (ICL) with large language models (LLMs) has emerged as the leading approach for this task (Maamari et al., 2024). Unlike traditional fine-tuning, ICL supplies instructions and a few demonstration examples directly in the model's input prompt, enabling models to generate SQL queries more effectively and efficiently (Rajkumar et al., 2022). Given that LLM performance is highly sensitive to the quality of these examples, creating optimal examples has become a critical area of research (Errica et al., 2024).

Current efforts to create demonstration examples for text-to-SQL rely on two primary approaches. The first involves manually annotating a small, fixed set of examples that are reused across queries (Pourreza & Rafiei, 2024). While straightforward, this approach often lacks flexibility and struggles with generalization. The second approach pre-generates a large pool of demonstration examples in advance[1], using retrieval techniques like similarity search to select relevant examples for each query. These examples, typically derived from training data, pair natural language questions with corresponding SQL queries (Poesia et al., 2022). The retrieval-based method using training data offers more adaptability and leads to performance improvements. Furthermore, Rajkumar et al. (2022); Chang & Fosler-Lussier (2023a) demonstrate that providing a small number of in-domain examples, *i.e.,* those from the same database as the test query, in the prompt results in better performance. This is because in-domain examples contain more relevant details and context, thereby

---

[1]For simplicity, we will refer to the large pool of demonstration examples as a "knowledge base" throughout the following text.

reducing the model's generalization burden. To address the challenge of acquiring in-domain data, Chang & Fosler-Lussier (2023b) propose synthesizing such data by adapting templates from external databases and populating them with columns and values from the target database, thereby lowering data acquisition costs. However, the absence of specific domain knowledge, such as the database schema or common query patterns, limits the model's ability to generate accurate SQL queries in cross-domain scenarios.

In practical settings, particularly within enterprise environments, multiple distinct business systems, such as flight booking and concert scheduling systems, often coexist (Panetto & Cecil, 2013), each with vastly different database structures (Panetto & Cecil, 2013). This variation complicates SQL query generation and demands greater adaptability and generalization from LLMs. We conducted extensive experiments comparing LLM performance on in-domain versus out-of-domain tasks for text-to-SQL, as detailed in Tables 1 and 3. The results indicate that LLMs perform significantly better on in-domain tasks, while their performance declines substantially when confronted with entirely different database structures. This underscores the current limitations of LLMs' generalization abilities in cross-domain scenarios. Consequently, we argue that for widely-used, high-accuracy tasks like text-to-SQL, effectively leveraging in-domain data is crucial for reducing the high generalization demands placed on LLMs, ultimately leading to improved performance and greater societal impact.

While the advantages of using in-domain data are evident, concerns may arise regarding the cost of acquiring such data for text-to-SQL tasks. Based on practical scenarios and our investigation, we provide two key reasons to demonstrate the abundance of in-domain data for text-to-SQL in real-world applications, which has been overlooked in previous work. **i)** According to the annual "Top 10 Programming Languages" report (IEEE Spectrum, 2023), SQL continues to dominate the "Jobs list", highlighting its widespread use across various enterprise roles, including Business Intelligence (BI) analysts, developers, database administrators (DBAs), product management, operations, compliance, and business strategy. This inevitably generates a large amount of in-domain data related to enterprise content and business. **ii)** Although LLMs excel at natural language tasks, their ability to generate complex SQL queries remains inferior to that of human experts. Consequently, LLMs are often used as assistive tools, generating initial SQL queries that users must review and modify. These correct SQL queries are often crafted by experts and accumulated as problems are solved. This valuable repository of correct query data, however, remains underutilized, even though it holds significant potential to enhance LLM performance.

Building on the above analysis and inspired by mistake notebooks, *i.e.,* a learning method commonly employed by students to track and learn from errors in exams or exercises (Yu et al., 2022), we propose a similar strategy for LLMs in the context of text-to-SQL tasks, shown in Fig. 1, referred to as LPE-SQL. In this approach, after each task, regardless of whether the generated SQL query is correct or erroneous, the results are logged into either a correct notebook or a mistake notebook. The correct notebook captures successful queries along with their reasoning paths, while the mistake notebook documents errors along with a reflection-generated tip designed to prevent the model from repeating similar mistakes in the future. By referencing these prior experiences, the model not only avoids previous errors but also reinforces its understanding by leveraging successful reasoning patterns. This iterative process allows the LLM to optimize its performance over time, improving both the accuracy of query generation and its ability to learn from accumulated experience. In enterprise settings, this feedback-driven mechanism holds the potential to significantly narrow the performance gap between LLMs and human experts in complex SQL generation, while also maximizing the value of existing data resources.

Our contributions can be summarized as follows:

- Departing from the conventional approach of using training sets as the knowledge base, we emphasize the value of in-domain data. We propose LPE-SQL, a continual learning method for text-to-SQL tasks that leverages in-domain data without requiring parameter fine-tuning, outperforming methods based on out-of-domain and synthetic in-domain data.

- Inspired by human learning strategies, we propose a novel knowledge base structure consisting of a correct notebook and a mistake notebook. By retrieving relevant entries from both notebooks during future tasks, the model is able to leverage prior experiences, improving SQL generation accuracy.

- Instead of simply pairing questions with their corresponding SQL queries, these notebooks are enriched with detailed reasoning paths and reflection-generated tips. Our experiments demonstrate that such high-information examples can further enhance model performance.
- Empirically, LPE-SQL demonstrates superior performance. Notably, the smaller Llama-3.1-70B-INT4 model, which utilizes the LPE-SQL method, outperforms the larger Llama-3.1-405B model, which employs SoTA techniques. The source code and all experimental results is open-source and available on GitHub[2].

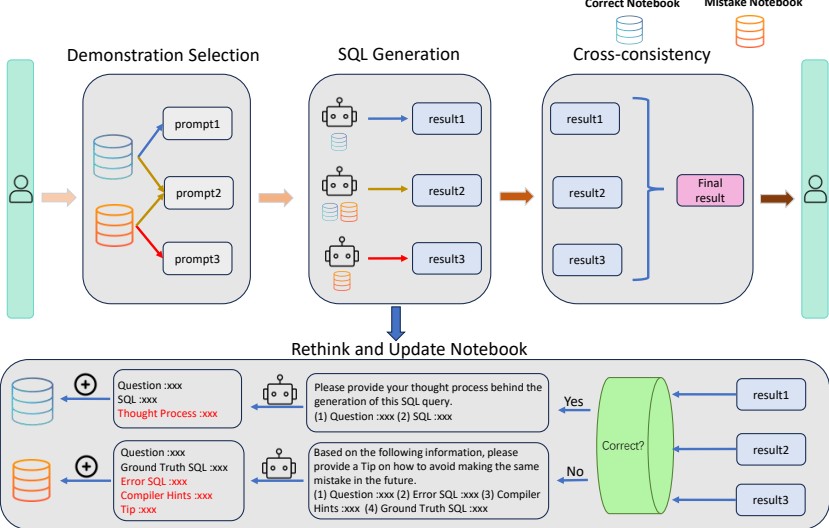

Figure 1: An overview of the proposed method including all four modules

## 2 RELATED WORK

SoTA Text-to-SQL approaches typically use a multi-module method consisting of demonstration selection, SQL generation, and a correction module (Maamari et al., 2024). Below, we discuss each module and relevant research.

**Demonstration Selection.** LLMs exhibit superior performance in Text-to-SQL tasks through in-context learning, where only a few examples are provided within input prompts. However, due to the high sensitivity of LLMs to prompt design, the success of in-context learning relies heavily on selecting appropriate examples (Errica et al., 2024). This module focuses on identifying the most effective examples from the knowledge base to mitigate this limitation. Many previous works have utilized the training set as a knowledge base, employing complex, multi-step retrieval strategies to select suitable examples (Zhang et al., 2023; Talaei et al., 2024; Xu et al., 2024; Wu et al., 2024). In contrast, Rajkumar et al. (2022) and Chang & Fosler-Lussier (2023a) found that performance can be significantly improved by using a small number of in-domain examples directly in the prompt, where question-SQL pairs correspond to the test database. Furthermore, Chang & Fosler-Lussier (2023b) argues that obtaining in-domain data is often challenging and proposes a hybrid knowledge base that combines synthetic in-domain data with out-of-domain examples. Nevertheless, both their experimental results and our findings (reported in Table 1) indicate that synthetic in-domain data is notably less effective than real in-domain data in enhancing performance. Unlike prior work, our approach utilizes a continuously expandable knowledge base that incorporates real in-domain data and out-of-domain data. We employ a straightforward, similarity-based matching retrieval method, which reduces the complexity of the approach while improving efficiency.

**SQL Generation.** The SQL generation phase often involves more than simply producing a candidate query from an input context. Prior works enhance this process by breaking it into multiple

---

[2]`https://anonymous.4open.science/r/LPE-SQL-3D58`

subtasks, solving them incrementally, and then merging the results (Talaei et al., 2024; Maamari & Mhedhbi, 2024; Pourreza & Rafiei, 2024). Common subtasks include schema selection, which encompasses both table and column selection. The objective here is to narrow the schema to include only the necessary tables and columns required for generating the SQL query. Another critical aspect is identifying query features and classifying them for targeted handling. For instance, Pourreza & Rafiei (2024) categorizes queries into three classes—easy, non-nested complex, and nested complex—tailoring prompts accordingly for each type. In our approach, we simplify this stage by directly instructing the LLM to generate SQL using straightforward prompts. This significantly reduces the complexity of the method and enhances the efficiency of SQL generation.

**Correction.** Two widely employed techniques at this stage are self-consistency (Wang et al., 2022) and self-debugging (Chen et al., 2023). For self-consistency, a notable example is MCS-SQL (Lee et al., 2024), which introduces a strategy that generates multiple prompts by varying the selection method and sequence of several demonstration examples, sampling multiple responses from LLMs to mitigate sensitivity. Their framework produces three distinct prompts per step, with each prompt executed 20 times by the LLM. In contrast, our approach executes the LLM only once per prompt, significantly reducing execution time and API costs, while presenting a greater challenge to our methodology. Regarding self-debugging, existing approaches typically rely on execution-based feedback (Andrew et al., 2024) or model-based feedback (Askari et al., 2024; Talaei et al., 2024) to correct generated SQL queries. For clarity, we integrate this strategy within the SQL generation module of our methodology.

# 3 METHODOLOGY

In this section, we introduce LPE-SQL, a continual learning framework for Text-to-SQL that utilizes real in-domain data without requiring parameter fine-tuning. It leverages human learning strategies, like mistake notebooks, to create a dynamic knowledge base for enhanced performance. Specifically, we give a detailed description of our proposed method, as shown in Fig. 1, the LPE-SQL method consists of four components: **i)** Demonstration Selection: demonstrations are selected based on question similarity from correct and mistake notebooks, with varying proportions prepended to the prompt for in-context learning. **ii)** SQL Generatation: these demonstrations, combined with the database schema and task-specific instructions as prompt, are used by the LLM to generate SQL queries. **iii)** Cross-consistency: multiple SQL results are compared to ensure consistency across different prompts, selecting the most stable result for increased robustness. **iv)** Rethink and Update Notebook: the generated SQL is evaluated against the ground truth; if correct, the model logs its thought process in the correct notebook. If incorrect, the model reflects on the failure and generates improvement tips for the mistake notebook, promoting continuous learning and refinement. We will discuss each component in detail.

## 3.1 DEMONSTRATION SELECTION

Given a few demonstration examples, LLMs can leverage them to generate SQL queries with a more standardized format and improved accuracy (Poesia et al., 2022). Demonstration examples selection plays a crucial role in few-shot learning, significantly impacting the performance of LLMs (Liu et al., 2021).

We adopt a hybrid strategy for demonstration selection, wherein we choose a total of $k$ demonstration examples from the knowledge base based on their embedding similarity to the test sample. The *correct rate* denotes the proportion of examples drawn from the correct notebook among the $k$ total examples. Specifically, we construct prompts by selecting examples from both the correct and mistake notebooks at three distinct ratios: *correct rate* = 1, where all $k$ examples originate from the correct notebook; *correct rate* = 0.5, featuring an equal distribution of examples from both notebooks; and *correct rate* = 0, where all examples are sourced from the mistake notebook. These examples, along with the database and the user's query, are integrated to form the final prompt. By varying the *correct rate*, we aim to thoroughly investigate and leverage the unique contributions of both the correct and mistake notebooks. The experimental analysis is reported in Section 5.

## 3.2 SQL GENERATATION

After the *Demonstration Selection* module, we utilize three distinct prompts. For each prompt, we employ a straightforward approach that enables the LLMs to directly generate a single SQL query. Following prevous work (Andrew et al., 2024), if an error message occurs during SQL execution, we use the error message as feedback. We take the database, user question, initial SQL, and error message as input, and ask LLMs to output the corrected SQL. This correction process is not illustrated in the SQL Generation module in Fig. 1.

## 3.3 CROSS-CONSISTENCY

Traditional self-consistency approaches in text-to-SQL tasks typically increase the randomness of LLM outputs by setting higher temperatures, thereby generating diverse SQL queries. These diverse queries are then aggregated through majority voting to determine the most consistent result (Lee et al., 2024). However, this method has notable drawbacks: higher temperatures can exacerbate model hallucinations (Renze & Guven, 2024), and the necessity for multiple API calls leads to increased costs in terms of token usage and runtime.

To address these limitations, we propose a novel cross-consistency method that utilizes diverse prompts generated during the *Demonstration Selection* and *SQL Generation* stages. After selecting demonstrations with varying correct rates (as described in Section 3.1), we construct distinct prompts. Each prompt instructs the LLM to generate a SQL query at a lower temperature, ensuring high quality and stability while reducing hallucinations. Importantly, only one API call is made per prompt, minimizing costs in terms of token usage and runtime. The resulting SQL queries are executed against the database, and their results are compared for consistency.

## 3.4 RETHINK AND UPDATE NOTEBOOK

The *Rethink and Update Notebook* module serves as a pivotal component of our LPE-SQL methodology, facilitating continuous learning through systematic evaluation and knowledge accumulation. This module comprises two essential processes: Rethink and Update Notebook.

**Rethink**: In the Rethink phase, the system evaluates the SQL queries produced by the *SQL Generation* module, comparing their execution results against those of the ground truth SQL to ensure alignment and accuracy. This evaluation mirrors the educational practice of maintaining a mistake notebook to enhance learning outcomes. Specifically, when execution results mismatch, the model performs reflective analysis to identify discrepancies and provide targeted improvement suggestions. Conversely, when match of both execution results, the model constructs a detailed reasoning process inspired by the "chain-of-thought" methodology (Wei et al., 2022). We expect this reasoning process to contribute to multi-step reasoning necessary for more complex tasks, thereby reducing the occurrence of hallucinations.

**Update Notebook**: The Update Notebook process operationalizes the insights gained from the Rethink phase by systematically integrating them into the relevant notebooks. This step involves updating the mistake notebook with reflective feedback and enhancing the correct notebook with validated reasoning chains, ensuring that accumulated knowledge is readily accessible for future tasks. By maintaining an up-to-date knowledge base of both successful and erroneous instances, the system leverages historical data to inform and improve future SQL query generation, fostering an environment of continual improvement and robustness.

## 4 EXPERIMENTAL SETUP

**Datasets and Evaluation Metrics.** We conduct our experiments on the BIRD dataset (Li et al., 2024), which is recognized as one of the most challenging benchmarks for Text-to-SQL tasks. BIRD consists of 12,751 unique question-SQL pairs derived from 95 comprehensive databases across 37 diverse domains, including education, healthcare, and more. Specifically designed to replicate real-world database scenarios, BIRD incorporates external knowledge and provides detailed descriptions of both databases and columns to address potential ambiguities in query generation. If not explicitly stated, all evaluations of LPE-SQL were conducted on the full development set.

To assess model performance, we utilize execution accuracy (EX), a metric that compares the execution results of the predicted SQL queries with those of the reference queries on the corresponding database instances. This approach measures the functional correctness of queries, accommodating variations in valid SQL formulations that yield the same results.

**Large Language Models.** We evaluate our approach using several leading LLMs, including Llama3.1-70B (Dubey et al., 2024), CodeLlama-34B (Roziere et al., 2023)[3], and GPT-3.5-turbo-0125 (GPT-3.5). Llama3.1 represents one of the most widely-used general-purpose open-source LLMs, while CodeLlama is a specialized variant of Llama2 (Touvron et al., 2023), further trained on code-specific datasets. GPT-3.5, a closed-source model, is among the most prominent LLMs in the field. Together, these models cover a broad spectrum of application scenarios. GPT-3.5 is accessed via OpenAI APIs, while the Llama3.1-70B and CodeLlama-34B model is downloaded from hugging face to perform experiments locally.

Due to the high computational restrictions and API cost of the most advanced models, such as Llama3.1-405B and GPT-4o, we were unable to include them in our experiments. However, it is noteworthy that the SoTA method by Maamari et al. (2024), which utilized Llama3.1-405B, achieved an EX score of 59.18 on a subset of the BIRD development set. In contrast, our approach, using the significantly smaller Llama3.1-70B model with INT4 quantization, achieved a notably higher score of 61.22 on the same evaluation set. This substantial performance improvement highlights the effectiveness of our method, with further analysis provided in the Section 5.

**Implementation Details.** For the initialization of the knowledge base, unless explicitly stated, we randomly selected 1,000 question-SQL pairs from the training set, leveraging our method to populate both the correct and mistake notebooks. For subsequent accumulation strategies of the knowledge base, we have two approaches: dynamically accumulating examples during the evaluation process following the LPE-SQL method, and not accumulating at all. Additionally, we drew inspiration from the work of Pourreza & Rafiei (2024) to create four manually annotated demonstration examples. Using the FAISS (Douze et al., 2024) library, we leveraged the similarities between the sentence embeddings[4] of the target and candidate questions to identify the most similar entries from both the correct and mistake notebooks, with the quantity determined by specific experimental settings. To minimize the randomness of the LLM outputs, we set the temperature to 0. All prompt templates are detailed in Appendix.

## 5 RESULTS

For thorough evaluation, the following research questions are addressed:

**RQ1: How important is real in-domain data?** We conducted a thorough analysis comparing real in-domain data with both out-domain data and synthetic in-domain data as knowledge base.

- **Real in-domain data vs. out-domain data:** We conducted experiments with Llama3.1-70B, CodeLlama-34B, and GPT-3.5 under two different knowledge base configurations: **i)** using randomly selected 1,000 question-SQL pairs from the training set, leveraging the notebooks generated by LPE-SQL to populate the knowledge base initialization method, without dynamically accumulating examples during the evaluation process, and **ii)** initializing the knowledge base as empty and only dynamically accumulating examples during the evaluation process. The results reported through the cross-consistency method, as shown in Table 1, demonstrate that the use of accumulated in-domain data outperforms the exclusive use of out-domain data by an average of 4.96% across these LLMs on the full BIRD development set. This improvement is primarily attributed to the alignment between the notebooks accumulated from in-domain data and the evaluation examples, which provides richer contextual information that aids the model in generating accurate responses. This indicates that real in-domain data is more effective than out-domain data in these scenarios.

---

[3]Due to the high GPU requirements of running Llama3.1-70B and CodeLlama-34B, we utilized the INT4 quantized and AWQ quantized versions, respectively, available on Hugging Face.

[4]The encoder model utilized is all-MiniLM-L6-v2, available on Hugging Face at `https://huggingface.co/sentence-transformers/all-MiniLM-L6-v2`.

- **Real in-domain data vs. synthetic in-domain data:** To the best of our knowledge, the only work that has explored synthetic in-domain data for text-to-SQL tasks is Chang & Fosler-Lussier (2023b). However, since the code for their data generation was not open-sourced, we could not replicate their exact results. Nevertheless, their experimental findings (Tables 1 and 2 in their paper) indicate that using only synthetic in-domain data leads to worse performance compared to solely using out-domain data across various LLMs and datasets. Combined with our findings from Table 1, it is clear that synthetic in-domain data fails to adequately represent real in-domain data.

Table 1: Performance comparison of out-of-domain and in-domain examples across different LLMs. Init method: Indicates the initialization method for the knowledge base, where "T" denotes the selection of 1,000 question-SQL pairs from the training set, with the notebook generated by LPE-SQL serving as the initial knowledge base. "-" indicates that the knowledge base is initialized as empty. Continuous accumulation: ✓ signifies that examples are continuously accumulated into the notebook during evaluation, while × indicates the opposite.

| Init method | Continuous accumulation | Model | | |
| --- | --- | --- | --- | --- |
| | | Llama 3.1-70B | CodeLlama-34B | GPT-3.5 |
| T | × | 59.45 | 48.76 | 55.48 |
| - | ✓ | 63.89 (**+4.44**) | 54.24 (**+5.48**) | 60.43 (**+4.95**) |

**RQ2: What is the impact of the correct notebook and mistake notebook on handling SQL generation with varying difficulty?** We conducted experiments in scenarios more aligned with the real world, specifically by initializing the knowledge base with a large amount of out-domain data and continuously adding new in-domain data. We selected demonstration examples from the correct and mistake notebooks, using three different selection ratios (detailed in Section 3.1), and analyzed SQL generation performance. Results are shown in Table 2.

Specifically, in Table 2, Llama-3.1 and CodeLlama consistently outperform the mistake notebook alone across most difficulty levels when using the correct notebook, particularly showing an average improvement of 4.49% on challenging tasks. This demonstrates that these models excel at learning the thought processes from the correct notebook to execute complex reasoning more effectively. In contrast, GPT-3.5 performs better using only the mistake notebook, indicating its strength in leveraging error experiences for reasoning tasks.

When both notebooks are used together, Llama-3.1 and CodeLlama show notable improvements in handling challenging tasks, benefiting from the integration of correct thought processes and error feedback, which helps reduce hallucinations and avoid repeated mistakes. Conversely, GPT-3.5 shows significant improvement in simple tasks, but its performance declines in moderate and challenging tasks compared to when it relies solely on the mistake notebook. This suggests that GPT-3.5 tends to learn from past errors more than from correct reasoning paths, particularly in more complex tasks, indicating a nuanced distinction in its learning process.

Through the analysis presented, we find that Llama-3.1 and CodeLlama achieve more consistent results, compared to GPT-3.5, which often exhibits an opposing trend. This discrepancy may arise from the more similar structure and training data of Llama-3.1 and CodeLlama, as opposed to GPT-3.5's differing characteristics. Consequently, these models demonstrate distinct learning patterns. Exploring these distinct learning patterns could be a valuable direction for future research.

**RQ3: What is the role of cross-consistency, and is it necessary?** From Table 2, it is evident that using both the correct and mistake notebooks in total yields superior results for Llama3.1-70B and GPT-3.5 compared to employing only the correct or mistake notebooks. However, CodeLlama-34B achieves its best performance solely with the correct notebook. Additionally, at more granular levels of difficulty, the differences among the models vary significantly depending on the *correct rate*. This suggests that the impact of the correct and mistake notebooks differs across various LLMs, likely due to variations in architecture, training data, or parameter settings, leading to distinct preferences in information processing. Consequently, without a clear understanding of the specific effects of the correct and mistake notebooks on a given LLM, achieving stable performance becomes challenging.

To address this inconsistency, we employed cross-consistency, and the results in Table 2 demonstrate that cross-consistency achieved either optimal results or results deviating by less than 1% from the best. However, it is important to note that implementing cross-consistency incurs additional time and API token costs. Exploring the applicability of the correct and mistake notebooks across different LLMs to fully leverage optimal combinations at various difficulty levels presents a promising avenue for future research aimed at enhancing performance while minimizing runtime and API token expenditures.

Table 2: EX scores on the full BIRD development set at different difficulty levels. Method: "CR-1", "CR-0.5", and "CR-0" correspond to *correct rate* = 1, *correct rate* = 0.5, and *correct rate* = 0, respectively, indicating the proportion of examples from the correct notebook. "Vote" refers to the results generated using cross-consistency. The highest and second scores for each model in each section are highlighted in red and blue, respectively.

| Model | Method | Difficulty level | | | Total |
| | | Simple | Moderate | Challenging | |
| --- | --- | --- | --- | --- | --- |
| Llama-3.1-70B | CR-1 | 70.92 | 56.47 | 47.59 | 64.34 |
| | CR-0.5 | 71.24 | 58.84 | 53.79 | 65.84 |
| | CR-0 | 69.62 | 57.54 | 44.83 | 63.62 |
| | Vote | 72.11 | 59.70 | 51.03 | 66.36 |
| CodeLlama-34B | CR-1 | 63.03 | 46.77 | 39.31 | 55.87 |
| | CR-0.5 | 62.16 | 45.26 | 42.07 | 55.15 |
| | CR-0 | 60.86 | 42.67 | 33.10 | 52.74 |
| | Vote | 62.81 | 45.26 | 37.93 | 55.15 |
| GPT-3.5 | CR-1 | 65.73 | 49.57 | 46.21 | 59.00 |
| | CR-0.5 | 69.41 | 50.86 | 46.21 | 61.60 |
| | CR-0 | 66.16 | 52.59 | 47.59 | 60.30 |
| | Vote | 68.22 | 51.94 | 44.83 | 61.08 |

**RQ4: How does performance improve as the number of entries accumulated in the notebook increases?** To answer RQ4 and comprehensively demonstrate the performance of LPE-SQL, we compared it with the current SoTA method in the BIRD benchmark (Maamari et al., 2024). For compare, we used the same evaluation set as theirs, which consisted of 10% of the entries from each database in the BIRD dev set. We conducted comparisons on Llama-3.1-70B using three different notebook configuration strategies, where all strategies initialized the notebook with 1,000 examples collected from the training set. The differences between the strategies are as follows: **i)** no further examples are accumulated during evaluation, **ii)** examples are continuously accumulated into the notebook during evaluation, and **iii)** to show performance improvement as the notebook grows, we accumulated the remaining examples from the BIRD dev set into the notebook, while continuously accumulating examples during evaluation. The results are shown in Table 3. Experimental results demonstrate that as the number of in-domain data entries accumulated in the notebook increases, Llama-3.1-70B demonstrates steady improvements in EX scores across various difficulty levels, with particularly notable gains at the challenging level, where performance doubles when utilizing a notebook rich in in-domain data. This strongly highlights the value of continual learning. Furthermore, we compared our approach with the SoTA method proposed by Maamari et al. (2024), which employs a comprehensive and iterative strategy. This method includes generating a candidate SQL query, then applying corrections through iterative re-generation based on database execution errors (Wang et al., 2018), revisions guided by database administrator instructions (Talaei et al., 2024), and model-based feedback akin to Reflexion (Shinn et al., 2024). It also incorporates self-consistency (Wang et al., 2022) to generate multiple responses and selects the most consistent result throughout the entire pipeline for augmentation, SQL generation, and SQL correction. In contrast, our approach, while only utilizing fundamental methods in each module, leverages continual learning and surpasses the SoTA method by 2.04%.

**RQ5: How do high-information examples enhance model performance?** We conducted two sets of experiments using the same experimental setup as in RQ2 on the Llama-3.1-70B model. In one setup, the examples in the notebook adhered to our proposed high-information methodology,

Table 3: The impact of increasing the scale of notebooks on the EX scores of the BIRD development subset across different difficulty levels. $E_{full}$ indicates that we accumulated the remaining examples from the BIRD development set into the notebook.

| Model | Init method | Continuous accumulation | Difficulty level | | | Total |
|---|---|---|---|---|---|---|
| | | | Simple | Moderate | Challenging | |
| Llama-3.1-70B | $T + E_{full}$ | ✓ | 67.90 | 53.70 | 50.00 | 61.22 |
| | $T$ | ✓ | 61.73 (**-6.17**) | 48.15 (**-5.55**) | 41.67 (**-8.33**) | 55.10 (**-6.12**) |
| | $T$ | ✗ | 60.49 (**-7.41**) | 42.59 (**-11.11**) | 25.00 (**-25.00**) | 51.02 (**-10.20**) |
| Llama-3.1-405B | SoTA(Maamari et al., 2024) | | - | - | - | 59.18 (**-2.04**) |

which included additional insights such as the thought process and relevant tips. In the other setup, the notebook contained only simple question-ground truth SQL pairs without any supplementary information. This setup aimed to highlight the impact of high-information examples on model performance.

As shown in Table 4, models with high-information examples consistently outperformed those trained with low-information examples across tasks of varying difficulty levels. Notably, the most significant improvement was observed in challenging tasks, where accuracy increased by 4.82%. Overall, the average accuracy across all difficulty levels reflected an improvement of 1.69% when utilizing high-information examples. These results underscore that providing the model with richer context and reasoning information significantly enhances its capability to handle more complex SQL generation tasks, particularly in challenging scenarios that require deeper reasoning and error correction.

Table 4: Performance comparison of Llama3.1-70B using low-information and high-information examples on the complete BIRD development set.

| Method | Difficulty level | | | Total |
|---|---|---|---|---|
| | Simple | Moderate | Challenging | |
| Low-information | 70.92 | 57.97 | 46.21 | 64.67 |
| High-information | 72.11 (**+1.19**) | 59.70 (**+1.73**) | 51.03 (**+4.82**) | 66.36 (**+1.69**) |

## 6 CONCLUSION

In this paper, we present LPE-SQL, a novel continual learning framework for Text-to-SQL tasks that utilizes real in-domain data without the need for parameter fine-tuning. Our approach draws inspiration from human learning strategies to create a dynamic knowledge base composed of a correct notebook and a mistake notebook. Extensive experiments conducted on the challenging BIRD dataset demonstrate the effectiveness of our method, as the smaller Llama-3.1-70B model outperforms the larger Llama-3.1-405B model using SoTA methodtechniques. Furthermore, we note that different large language models display varying learning patterns with the correct and mistake notebooks. Leveraging these distinct patterns could further enhance performance, presenting a valuable area for exploration.

## 7 LIMITATIONS

This work has several limitations and areas for improvement. Since advancing the SoTA was not our primary goal and due to the high cost of APIs for advanced models like GPT-4o, we did not evaluate our approach with the most powerful LLMs. Additionally, we did not incorporate schema linking, a common technique for recognizing tables and columns, nor did we use methods like high-temperature self-consistency to generate multiple candidate SQL queries from a single prompt. As a result, the full potential of our approach remain unexplored. We believe that our continuous learning framework for integrating in-domain data into the correct and mistake notebooks is applicable not only to Text-to-SQL tasks but also to other reasoning tasks, such as mathematical reasoning, which we will continue to explore.

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
