# LEVERAGING PRIOR EXPERIENCE: AN EXPANDABLE AUXILIARY KNOWLEDGE BASE FOR TEXT-TO-SQL (SUPPLEMENTARY)

## A  PROMPT TEMPLATES

This section introduces the prompt templates used in LPE-SQL, categorized into four types: the template for generically generating SQL queries (List 1), the template for generating the corresponding thought process based on the SQL query (List 2), the template for generating tips based on the incorrect SQL and the ground truth SQL (List 3), and the template for re-generating the SQL using error information from SQL execution (List 4). For demonstration purposes, we use a scenario that combines both the correct and mistake notebooks ($correct\ rate = 0.5$).

```
# For your reference, here are some examples of Questions, sql queries,
and thought processes related to the Question you're working with
{Example2}

# Below are examples of mistakes you've made before that are similar to
the question you're about to tackle, so please refer to not making
the same mistake!
{Example1}
{Example2}

# Schema of the database:
{Database Schema}

-- Using valid SQLite and understanding Hint, answer the following
    questions for the tables provided above.
-- {Question}
-- {External Knowledge}

Generate the SQLite for the above question after thinking step by step:

In your response, you do not need to mention your intermediate steps.
    Do not include any comments in your response.
    Do not need to start with the symbol ```
    Your SQL code should be concise and efficient.
    You only need to return the result SQLite SQL code
    start from SELECT
```

Listing 1: The template for generically generating SQL queries.

```
# Schema of the database:
{database_schema}

# Question:
{Question}

# External Knowledge :
{External Knowledge}

# You just generated the following SQL:
{SQL Query}
```

```
Now, please provide your thought process behind the generation of this
SQL query. Your explanation should be concise and efficient, focusing
on the key reasoning steps.
```

Listing 2: The template for re-generating an SQL query based on error feedback from SQL execution.

```
# Schema of the database:
{Database Schema}

# Question:
{Question}

# External Knowledge :
{External Knowledge}

# Error SQL Query:
{Error SQL Query}

# Error information:
{Error}

# SQL after Reflection:
{SQL after Reflection}

# Ground Truth SQL:
{Ground Truth SQL}

Error SQL Query is the result you generate the first time and SQL after
Reflection is the result you generate again based on the Error
information returned by the compiler knowing that the first generated
result was wrong. Now that both results are known to be wrong, I am
providing Ground Truth SQL for your reference, please think carefully
about why your first two results were not correct, please provide a
Tip on how to avoid making the same mistake in the future. Note that
you only need to return the Tip. Please return in the following format:
# Tip:
```

Listing 3: The template for generating tips based on the incorrect SQL and the ground truth SQL.

```
# For your reference, here are some examples of Questions, sql queries,
and thought processes related to the Question you're working with
{Example2}

# Below are examples of mistakes you've made before that are similar to
the question you're about to tackle, so please refer to not making
the same mistake!
{Example1}
{Example2}

# Schema of the database:
{Database Schema}

# Question:
{Question}

# External Knowledge :
{External Knowledge}

# SQL Query:
{SQL Query}

# Error:
{Error}
```

```
Reflect on the error encountered in the SQL query and provide a corrected
SQL query.

In your response, you do not need to mention your intermediate steps.
    Do not include any comments in your response.
    Do not need to start with the symbol ```
    Your SQL code should be concise and efficient.
    You only need to return the result SQLite SQL code
    start from SELECT
```

Listing 4: The template for generating a thought process corresponding to the SQL query.

## B    REASONING PIPELINE

To clarify the proposed LPE-SQL method, we provide a summary of the reasoning pipeline in Algorithm Tables 1 and 2. Algorithm Table 1 outlines the reasoning process for a single *correct rate* setting, whereas Algorithm Table 2 demonstrates the reasoning process using the cross-consistency method across various *correct rate* settings. The complete source code is available in *src/gpt_request.py*.

---

**Algorithm 1** Main Pipeline of Single Reasoning for LPE-SQL

---

**Input:** Initialization of the knowledge base $KG$, correct rate $CR$, number of demonstration examples $k$, Question $Q$, External Knowledge $EK$, database path $db\_path$, ground truth SQL $GT$.
**Output:** SQL query.

 1: Initialize a retriever ($CR$, $KG$) to retrieve and update data in $KG$.
 2: Demonstration example $E \leftarrow$ retriever.get_example($Q$)
 3: Prompt $\leftarrow$ generate_prompt_common_sql($Q$, $E$, $EK$)
 4: SQL query $\leftarrow$ LLM(Prompt)
 5: Prompt $\leftarrow$ generate_prompt_thought_process($Q$, $EK$, SQL query)
 6: Thought process $\leftarrow$ LLM(Prompt)
 7: _, error $\leftarrow$ execute_sql(SQL query, $db\_path$)
 8: **if** error $\neq$ None **then**
 9:     Prompt $\leftarrow$ generate_prompt_reflection_sql($E$, $Q$, $EK$, SQL query, error)
10:     Reflectioned SQL query $\leftarrow$ LLM(Prompt)
11: **end if**
12: Predicted SQL $\leftarrow$ **if** New SQL query $\neq$ None **then** New SQL query **else** SQL query
13: res $\leftarrow$ execute_compare(Predicted SQL, $GT$)
14: **if** res == 0 **then**
15:     Prompt $\leftarrow$ generate_prompt_reflection_tip($Q$, $EK$, SQL query, error, Reflectioned SQL query, $GT$)
16:     Tip $\leftarrow$ LLM(Prompt)
17:     $KG \leftarrow$ retriever.add_to_mistake_notebook($Q$, $EK$, SQL query, error, Reflectioned SQL query, $GT$, Tip)
18: **else**
19:     $KG \leftarrow$ retriever.add_to_correct_notebook($Q$, $EK$, Predicted SQL, Thought process)
20: **end if**
21: Obtain the predicted SQL query and updated knowledge base $KG$.

---

**Algorithm 2** Main Pipeline of Cross-Consistency Reasoning for LPE-SQL

---

**Input:** Initialization of all knowledge bases $KG\_list$, list of all correct rates $CR\_list$, number of demonstration examples $k$, Question $Q$, External Knowledge $EK$, database path $db\_path$, ground truth SQL $GT$.
**Output:** Final SQL query.

 1: Initialize a list $sql\_list$ to store all generated SQL queries.
 2: **for** each $CR$, $KG$ in $CR\_list$ and $KG\_list$ **do**
 3:     Use Algorithm 1 to obtain the SQL query based on the current $CR$ and $KG$, and save it into $sql\_list$.
 4: **end for**
 5: Compare the execution results of all SQL queries in $sql\_list$, and select the SQL query with the most consistent results as the final SQL query.
 6: Obtain the final SQL query and all updated knowledge base.

---

# C   MORE RESULTS

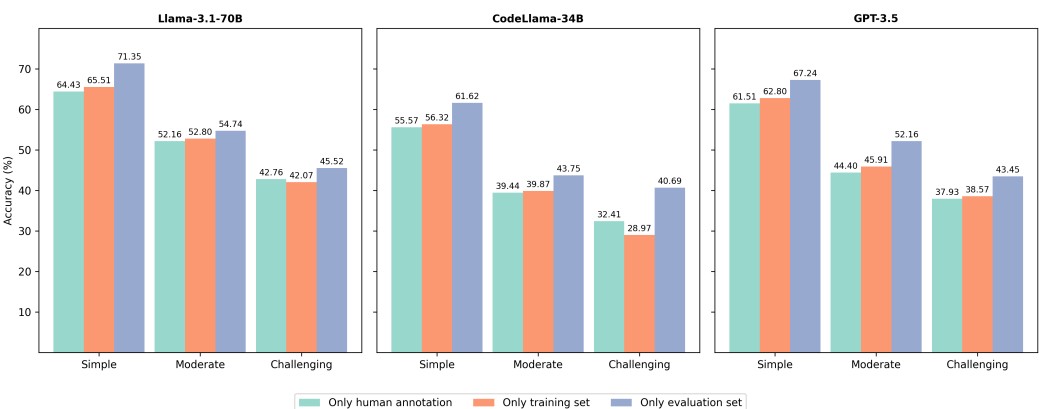

Figure 1: EX scores across problems of varying difficulty levels from the BIRD development dataset using different methods.

In Fig. 1, we present a comparison of different methods applied to problems of varying difficulty levels from the BIRD development dataset. These methods include: **i)** using only manually annotated examples based on Pourreza & Rafiei (2024), **ii)** using 1000 examples collected from the training set, and **iii)** using examples dynamically accumulated during evaluation via the LPE-SQL method. The first two methods are commonly used in few-shot learning as knowledge base, while the third method is introduced in our LPE-SQL approach. Consequently, a detailed examination of our approach, along with a comparison to other methods, by analyzing performance across problems of varying difficulty at a more granular level, provides valuable insights.

**Using the training set as a knowledge base does not significantly outperform carefully designed fixed examples.** Across all three different LLMs tested in the experiment, using the training set as a knowledge base provided a slight performance improvement—around 1%—over manually annotated examples for tasks of simple and moderate difficulty. However, at the challenging difficulty level, both Llama-3.1-70B and CodeLlama-34B showed consistent performance drops, with CodeLlama-34B experiencing a decline of 3.44%. These observations indicate that there is no significant difference in performance between these two methods.

**In-domain data accumulation leads to comprehensive improvements.** Compared to both using the training set as a knowledge base and relying on carefully designed fixed examples, continuously accumulating domain-specific data during evaluation results in significant improvements across various difficulty levels. At the simple, moderate, and challenging levels, applying the *evaluation-only* method with different LLMs achieves at least 4.44%, 1.94%, and 2.76% improvements over the other two methods, respectively. The maximum observed improvements reach 6.92%, 7.76%, and 11.72%, underscoring the effectiveness of this approach.

## REFERENCES

Mohammadreza Pourreza and Davood Rafiei. Din-sql: Decomposed in-context learning of text-to-sql with self-correction. *Advances in Neural Information Processing Systems*, 36, 2024.