# OpenReview forum: "Leveraging Prior Experience: An Expandable Auxiliary Knowledge Base for Text-to-SQL"
_ICLR.cc/2025/Conference — ICLR 2025 Conference Withdrawn Submission_

### Official Review · Reviewer_JaqR · 2024-10-24

**Soundness:** 2
**Presentation:** 3
**Contribution:** 2
**Rating:** 3
**Confidence:** 5

**Summary:**

This paper proposes a framework to help LLMs better perform on the text-to-SQL task.
The proposed LPE-SQL framework comprises several parts, retrieving relevant entries or the datapoints in the training or the evaluation process, incorporating various prompts with various entries or demonstrations to check cross-consistency, in which the correct ones will be placed into correct notebook, while the incorrect ones will be sent to mistake notebook. Also, the notebook will be updated through the rethinking process, where the LLM will incorporate the thought process, error SQL, compiler hints, etc., into the notebook.

The authors have studied the various questions including the effects of how to select entries for the notebook, whether to use real in-domain data or the out-of-domain data, cross-consistency, etc., to help better understand the contribution of each component.

**Strengths:**

- The authors have proposed a framework, which leverages the demonstration process to enhance models' text-to-SQL performance.
- The authors have studied various questions, trying to understand the contributions of different components in their proposed framework.

**Weaknesses:**

- There is only one text-to-SQL dataset studied, the BIRD dataset, which limits the breath of this study. Even in the text-to-SQL domain, there are other text-to-SQL datasets there, including the single-domain ones such as ATIS, Advising, as well as previous cross-domain one Spider and a bunch of its variants. Moreover, as observed by researchers in practice, there are noises in the BIRD dataset [1], and the dataset itself is constantly updated. Considering that the authors claim that `advancing the SoTA was not our primary goal`, I would suggest the authors to include more text-to-SQL datasets into their studies.
- I understand that because of limited model access, the authors did not build their study upon the advanced models like GPT-4o. However, if the authors examine the BIRD benchmark leaderboard (https://bird-bench.github.io/), there is information on the size of each method, and their corresponding performance. Just to name a few here: ExSL + granite-34b-code at the scale of 34B, achieves 67.47 and 70.37 on the BIRD dev and test set, respectively, which surpasses the performance reported in this paper on BIRD benchmark, with a Llama based 70B model (The paper seems cryptic about the best performance they achieve, in Table 4, it seems that the best performance they achieve is 66.36 on the Bird dev set, and they did not report any scores on BIRD test set.) In addition, there seems to be no baseline methods here for comparison, which undermines the contributions of this work. By baselines, I mean not only the representative methods proposed by the previous works but also baselines such as supervised fine-tuning on those examples in the notebook, and directly feeding those datapoints from the notebook to the model, etc. Such comparison would enhance readers' understanding of the method in terms of the performance lower bound and upper bound.
- The RQs are shallow, and the answer to those RQs are not satisfying. For instance, for cross-consistency, and performance improvement as the number of entries accumulate in the notebook, I do not think they are that intriguing in terms of novelty, as readers may form a natural intuition on the outcomes. The authors may spend less space discussing on these topics but propose more interesting items to study. In terms of the answer, the analysis for RQ2 only scratches the surface. For instance, claims such as `GPT-3.5 performs better using only the mistake notebook, indicating its strength in leveraging error experiences for reasoning tasks` is neither well explained, nor grounded by concrete examples. Such shallow claims include ` This suggests that GPT-3.5 tends to learn from past errors more than from correct reasoning paths, particularly in more complex tasks, indicating a nuanced distinction in its learning process`, etc. These claims sound ungrounded, and the final sentence for this RQ, `Exploring these distinct learning patterns could be a valuable direction for future research.` diverges from the main topic of this paper, and fail to deliver concrete and useful information to inspire future studies.


### Typos:
- Please capitalize `sql`, e.g., on line 022, `efficient sql` --> `efficient SQL`.



## References
[1] Wretblad, Niklas, et al. "Understanding the Effects of Noise in Text-to-SQL: An Examination of the BIRD-Bench Benchmark." arXiv preprint arXiv:2402.12243 (2024).

**Questions:**

- It seems that RQ4 is addressing the effects of `k` but I am still confused on how many examples will be fed to the LLM eventually at its inference time?

---

### Official Review · Reviewer_fNsa · 2024-10-30

**Soundness:** 2
**Presentation:** 3
**Contribution:** 2
**Rating:** 3
**Confidence:** 4

**Summary:**

This paper presents LPE-SQL, which enhances large language models (LLMs) by enabling continual learning without parameter fine-tuning.  The proposed LPE-SQL introduces four modules: 1) demonstration selection, 2) SQL generation, 3) Cross-consistency (voting) correction to obtain the final result, and 4) Rethink and Update Notebook for knowledge accumulation.  More specifically, it simply records successful and failed examples, along with their reasoning or reflections, to build in-domain instances. By retrieving similar examples, it provides LLMs with relevant context for generating better outputs, while also recording the current one. The authors highlight the importance of continual learning through extensive ablation studies on the BIRD dataset.

**Strengths:**

1. Within the framework of continuous learning, the author introduced in-domain examples to improve the performance of LLM in text-to-SQL.
2. The authors add reasoning paths and reflection prompts to the examples to provide additional guidance for LLM, further improving the performance of LLM.

**Weaknesses:**

1. Technical contributions of the paper：
1.1 The proposed LPE-SQL seems just a combination of several modules without sufficient technical contribution.  Especially, the authors claim the Rethink and Update Notebook module is a solution for continual learning.  However, the solution feels common without much learning mechanism.
1.2 The cross-consistency correction module seems unnecessary.  The cross-consistency correction module just adopts the voting mechanism to select the most appropriate SQL statement based on different prompts.  However, in Table 2, the performance of CR-0.5 and vote (cross-consistency) appears to be similar, achieving either optimal results or results deviating by less than 1% from the best. It seems that CR-0.5 is more cost-effective than the voting mechanism (or in the proposed LPE-SQL) because CR-0.5 requires fewer API tokens.  Hence, the results in Table 2 seem to conclude that cross-consistency correction is not necessary.
2. Completeness of the paper
2.1 The experiment is only conducted on the BIRD dev.  The generalization of the proposed method is unclear.
2.2 In the Demonstration Selection stage, it remains unclear whether the method adopted by the author is better than other existing methods, e.g., MCS-SQL (https://arxiv.org/pdf/2405.07467), DIN-SQL (https://arxiv.org/pdf/2304.11015), and ODIS (https://arxiv.org/pdf/2310.06302).
2.3 The evaluation of the Demonstration Selection shown in Table 2 is not complete.  The reviewer is expected to see the performance when CR is set to more values, e.g., 0.2, 0.4, 0.6, 0.8, etc.
2.4 Lack of sensitivity analysis of parameter k (the number of samples in Demonstration Selection): The reviewer wants to see the impact of the number of the selected samples to the performance.
2.5 When continuously adding samples, will the order of evaluation affect the results?

**Questions:**

1. If the notebook capacity is large enough, should we consider pruning? What is the performance?
2. See the questions in Weakness.

---

### Official Review · Reviewer_wLR4 · 2024-11-03

**Soundness:** 2
**Presentation:** 3
**Contribution:** 2
**Rating:** 5
**Confidence:** 3

**Summary:**

This paper focuses on the well-studied text-to-SQL task, where the goal is to convert natural language instructions to SQL commands. Previous studies show that in-domain data used as in-context examples significantly improves performance compared to fixed examples or synthetic data. The paper argues that real-world text-to-SQL data is plentiful and could be leveraged for in-domain examples. They propose LPE-SQL, a framework inspired by human “mistake notebooks,” where both correct and incorrect queries are recorded for continual learning. Experiments on the BIRD dataset with three LLM backbones demonstrate the approach’s effectiveness.

**Strengths:**

- **Originality**
    - LPE-SQL introduces a novel continual learning approach to the text-to-SQL task by utilizing a dynamically updated knowledge base containing correct and mistake notebooks.
- **Quality and Clarity**:
    - The paper is generally easy to follow and has informative figures.
    - The research questions posed in Section 5 are well-defined and helpful for better understanding the paper.
    - The experiments are thorough, especially given the variety of difficulty levels and the use of both in-domain and out-of-domain data.

**Weaknesses:**

- Minor writing problem for Section 1: While the notebook mechanism is well-explained, the role of “continual learning” isn’t clearly integrated into the introduction or methodology. Indeed, the word “continual” is not mentioned once in the introduction besides the bullet points.
- The paper briefly notes that LPE-SQL performs competitively with or outperforms some SoTA methods; however, no numbers are included in the paper for this comparison.
Moreover, it would be better to test on multiple benchmarks beyond BIRD to help contextualize the approach’s effectiveness.

**Questions:**

- My main concern is whether ICLR is the right venue for this paper. Given the paper’s focus on practical improvements in SQL query generation and data utilization, it may be better suited to an applied AI conference with an emphasis on NLP systems or LLMs in industry contexts.

---

### Official Review · Reviewer_Peu9 · 2024-11-11

**Soundness:** 2
**Presentation:** 2
**Contribution:** 2
**Rating:** 5
**Confidence:** 4

**Summary:**

This paper investigates LLMs for text-to-SQL with continual learning. The method is called LPE-SQL.

The main idea is to maintain a mistake notebooks  to log the errors along the reasoning paths with reflection-based tips.
The logs can be retrieved accordingly for a new input query to improve the text-to-SQL generation quality.

Experimental results on BIRD show that with the proposed method, a LLaMa-3.1-70B-INT4 model can outperforms the performance of the LLaMa-3.1-405B SOTA model.

**Strengths:**

1. The proposed method is novel and reasonable.  The main idea is to use a knowledge base full of correct and wrong reasoning paths to better guide Text-to-SQL generation.

2. The general writing is clear and easy-to-understand.

3. On a subset of BIRD development,  the proposed method based on a LLaMa-3.1-70B-INT4 model  give better results than the SOTA model based on LLaMa-3.1-405B (61.22 v.s. 59.18).

**Weaknesses:**

1. One major problem is that where the performance gain is from  is unclear.  It is unknown how many demonstrations are used in the previous SOTA model, but for this model, it uses more than 1000 examples as the knowledge base.

2. Second,  this paper does not fully confirm the contribution of the proposed mistake notebook.    For comparing with the SOTA model,  the experimental results are based on a subset of BIRD.  But for the ablation experimental results in Table 2, they are done on the full set of the BIRD.  Can you also show the ablation studies on the same subset of BIRD for comparing SOTA and this can show that the mistake notebook is necessary.

	In addition, the SOTA method of (Maamari et al. 2024) does not seem to direct use the remaining development set to construct the prompts. However, in Table 3,  for the continual learning, it also includes E_{full} as the knowledge base, which seems to be unfair.  Can you also show the performance of  the init method with only E_{full} in Table 3.

3.  The cross-consistency method seems not to be necessary.  It does not outperform the CR-0.5 method in general in Table 5.

4. The overall evaluations are done in one dataset (BIRD). The authors should consider more related datasets.

**Questions:**

Please see above in Weaknesses.

---

### Note · Authors · 2024-11-20

I have read and agree with the venue's withdrawal policy on behalf of myself and my co-authors.